# Humoral and Cellular Immune Responses Against SARS-CoV-2 Following COVID-19 Vaccination in Older Adults: A Systematic Review

**DOI:** 10.3390/vaccines13080852

**Published:** 2025-08-12

**Authors:** Ruth Angélica Rojas-De la Cruz, Janeth M. Flores-Córdova, Cielo Cinthya Calderon-Hernandez, Nelson Luis Cahuapaza-Gutierrez, Nino Arturo Ccallalli-Ruiz, Fernando M. Runzer-Colmenares

**Affiliations:** 1Facultad de Ciencias de la Salud, Carrera de Medicina Humana, Universidad Científica del Sur, Lima 15067, Peru; 100077368@cientifica.edu.pe (R.A.R.-D.l.C.); 100076390@cientifica.edu.pe (J.M.F.-C.); 100052073@cientifica.edu.pe (C.C.C.-H.); 100065659@cientifica.edu.pe (N.L.C.-G.); 2CHANGE Research Working Group, Universidad Científica del Sur, Lima 15067, Peru; nccallallir1@upao.edu.pe; 3Research Department, N y C—Center of Research and Medical Excellence (CRME), Lima 15067, Peru; 4Facultad de Ciencias de la Salud, Carrera de Medicina Humana, Universidad Privada Antenor Orrego, Piura 20009, Peru

**Keywords:** COVID-19 vaccine, immunosenescence, cellular immunity, humoral immunity, older adults (source: MeSH NLM)

## Abstract

Background: Evidence on the humoral and cellular immune responses to SARS-CoV-2 following COVID-19 vaccination in older adults is warranted. Aims: To synthesize and analyze the current evidence on humoral and cellular immune responses to both standard and booster COVID-19 vaccination in individuals aged 60 years and older. Methods: Clinical trials and observational studies were included. Reviews, case series, letters to the editor, and similar publications were excluded. A selective literature search was conducted in the following databases: PubMed, Scopus, EMBASE, and Web of Science. The risk of bias and methodological quality of the included studies were assessed using the Newcastle–Ottawa Scale (NOS) and the Risk of Bias 2.0 (RoB 2) tool. Statistical analysis was conducted using Stata version 18 and Review Manager version 5.4.1. Results: Thirteen studies were included: eleven observational studies and two randomized clinical trials, evaluating humoral and cellular immune responses in 782 older adults. Messenger RNA vaccines were the most administered, particularly Pfizer-BioNTech (76.9%) and Moderna mRNA-1273 (23%). In most cases, immune responses were assessed after the second dose and booster doses. Most studies (61.5%) reported increased IgG titers specific to the SARS-CoV-2 Spike protein, while 23.1% reported a decrease. Regarding cellular immunity, 46.2% of the studies reported low interferon-gamma (IFN-γ) levels post-vaccination, whereas 38.5% showed increases. These findings highlight the need for tailored vaccination strategies to address emerging variants, particularly in vulnerable populations such as older adults. Conclusions: In older adults receiving COVID-19 vaccination, humoral immunity tends to increase, whereas cellular responses are frequently diminished, reflecting age-related immunosenescence that may limit the durability and breadth of protection following vaccination in older adults.

## 1. Introduction

COVID-19 is a disease that primarily affects the upper respiratory tract and is caused by SARS-CoV-2, a single-stranded RNA virus of the Betacoronavirus genus [1,2]. Since its emergence, COVID-19 has spread rapidly, with high morbidity and mortality rates, particularly among older adults [3]. To address this threat, the U.S. Food and Drug Administration (FDA) granted emergency authorization for several vaccines, a key public-health strategy to reduce hospitalizations and deaths [4]. These vaccines include inactivated and live-attenuated formulations, protein subunits, viral vectors, RNA and DNA platforms, and virus-like particles, all designed to induce long-term immunity without full exposure to the disease [4,5].

During the COVID-19 pandemic, adults aged 60 years and older were among the most severely affected, and were therefore prioritized in vaccination schedules [6]. However, vaccines that are highly effective in younger populations may elicit weaker immunity in older individuals because of progressive immune decline [7]. This age-related dysfunction, known as immunosenescence, affects both the innate and adaptive arms of the immune system: it is multifactorial and dynamic, characterized by reduced antigen responsiveness, an expansion of memory T cells, and low-grade chronic inflammation [8,9]. Immunosenescence contributes to an age-related reduction in the initial response to vaccines and is also associated with a more rapid decline in antibody levels following vaccination. Furthermore, it may negatively affect the generation of antibodies in response to viral infections and vaccines [3,10].

Recent studies have documented diminished humoral and cellular responses in older adults following COVID-19 vaccination [11]. Other investigations show that primary vaccination induces lower immune responses in older compared with younger adults, although this immunogenicity gap may narrow with additional doses [12]. For example, booster doses of the Pfizer-BioNTech vaccine have been shown to increase IgG titers and enhance T-cell responses in older adults, and a separate study demonstrated boosted antibody- and T-cell-mediated immunity in individuals aged 60 years and older, after a fourth Pfizer-BioNTech dose [13,14]. However, the reduced response to vaccines in older adults is not always due exclusively to immunosenescence, but may also be influenced by other factors such as comorbidities, nutritional status, and baseline immune health.

Despite these findings, data on the long-term persistence of humoral and cellular immunity in older adults after COVID-19 immunization remain limited. Given that the COVID-19 pandemic emerged only five years ago, long-term studies are inherently scarce. However, key questions remain regarding the durability of protection and the immunogenicity of booster doses in this vulnerable population. Consequently, synthesizing the current evidence is essential. The primary objective of the present study was to review, analyze, and integrate the available data on humoral and cellular immune responses in adults aged 60 years and older.

## 2. Materials and Methods

### 2.1. Protocol and Registration

The study protocol was developed in accordance with the “Preferred Reporting Items for Systematic Reviews and Meta-Analyses for Protocols” PRISMA-P 2015 statement [15]. The review protocol was registered in the “International Prospective Register of Systematic Reviews” PROSPERO under registration number CRD420251042535. This systematic review was conducted following the PRISMA 2020 guidelines [16].

### 2.2. Objectives

The primary objective of this study was to review, analyze, and synthesize the current evidence on humoral and cellular immune responses induced by COVID-19 vaccines in older adults. The population of interest included individuals aged 60 years and older. The intervention comprised all types of COVID-19 vaccines, regardless of the number of doses administered (1st, 2nd, or ≥3rd) and the platform used (inactivated, viral vector, nucleic acid, or protein subunit vaccines). Comparators included placebo, saline solutions, or unvaccinated control groups. The primary outcome was the evaluation of immune response, both cellular and humoral. Cellular immune response was assessed through T-cell activation, including the secretion of cytokines such as tumor necrosis factor-alpha (TNF-α) and IFN-γ. Humoral immune response was measured through the presence of SARS-CoV-2–specific IgG antibodies, particularly those targeting the Spike glycoprotein (anti-S).

### 2.3. Eligibility Criteria

Eligibility criteria were based on the previously formulated PICO research question. Included studies were phase ≥ 3 clinical trials and observational studies (cohort and case-control designs). Excluded studies were phase 1/2 clinical trials, cross-sectional observational studies, case reports, case series, letters to the editor, editorials, clinical images, letters, comments, notes, correspondences, short communications, brief reports, conference abstracts, narrative reviews, systematic reviews, books, book chapters, news articles, and opinion pieces. Studies reporting different outcomes, populations, or types of interventions were also excluded.

### 2.4. Information Sources

A preliminary search was conducted on 15 April 2025, and updated on 5 May 2025. The final search presented in this review was performed on 16 May 2025. A selective literature search was carried out in the following electronic databases: PubMed, Scopus, EMBASE, and Web of Science. In addition, reference lists of included studies were manually screened to ensure comprehensive search.

### 2.5. Search Strategy

The search strategy was developed using Medical Subject Headings (MeSH) from the National Library of Medicine (NLM), including the terms: “COVID-19 Vaccines,” “Immunosenescence,” “Immunity, Humoral,” “Cellular Immunity,” and “Older adults,” combined with Boolean operators AND and OR. No publication date restrictions were applied. The search was limited to studies published in English. The detailed search strategy for each database is provided in the Appendix A.

### 2.6. Study Selection Process

All references were downloaded and exported to the Rayyan QCRI web platform to remove duplicates. Two authors (C.C.C.-H., R.A.R.-D.l.C) independently screened titles and abstracts of potentially eligible studies. Full-text articles were then assessed to determine final eligibility. Any disagreements were resolved by a third author (NLCG).

### 2.7. Data Extraction Process

Data was extracted into a Microsoft Excel spreadsheet prepared in advance by the authors. Key study characteristics were extracted, including first author, year of publication, country, study design, study period, age group, vaccine type, number of doses, immune response (cellular and humoral), main findings, and conclusions. Disagreements were resolved in consultation with the other authors. No attempts were made to contact original study authors for missing data.

### 2.8. Risk of Bias and Quality Assessment

To assess the risk of bias in randomized controlled trials, the Cochrane RoB 2 tool was used [17]. This tool evaluates five domains where bias can be introduced: the randomization process, deviations from intended interventions, missing outcome data, measurement of the outcome, selection of the reported result, and overall bias. Each domain is judged as having low risk, some concerns, or high risk of bias. For observational studies, the NOS was applied to assess the risk of bias in both case-control and cohort studies [18]. This tool evaluates studies based on three predefined domains: selection, comparability, and exposure (for case-control studies) or outcome (for cohort studies).

### 2.9. Statistical Analysis

A high degree of heterogeneity was identified among the included studies. Consequently, the synthesis of findings was limited, and stratified in accordance with methodological guidelines for systematic reviews. The process included a comprehensive literature search, study selection, full-text review, data extraction, and risk-of-bias assessment. Due to substantial differences across studies, such as the characteristics of the populations analyzed, definitions of outcomes (humoral and cellular responses), and follow-up periods valid statistical meta-analysis was not feasible. An attempt was made to harmonize the data by grouping studies with comparable outcomes and similar reporting formats; however, significant differences persisted, which would have compromised the validity and interpretation of quantitative synthesis. Therefore, a structured narrative synthesis was conducted, presenting findings by individual studies (Appendix A).

## 3. Results

### 3.1. Selection of Studies

A selective literature search was conducted in four electronic databases: PubMed, Scopus, EMBASE, and Web of Science, identifying a total of 1684 records. After removing duplicates, 1173 manuscripts remained. An initial screening by title and abstract identified 84 potentially eligible studies. These were then assessed in full text, with 58 excluded for not meeting the inclusion criteria, resulting in 26 selected studies. To ensure comprehensive search, the reference lists of these 26 studies were reviewed, yielding 2 additional relevant articles. In total, 26 studies were analyzed, of which 15 were excluded due to reporting other outcomes, including populations under 60 years of age, providing incomplete numerical data, or being of a different study type. Ultimately, 13 studies were included in the present review. Figure A1 describes the study selection process.

### 3.2. Characteristics of the Included Studies

This review included 13 studies; 11 cohort studies and 2 randomized controlled trials published between 2021 and 2025. Humoral and cellular immune responses were analyzed in 782 older adults (≥60 years). Most participants received messenger RNA vaccines, primarily Pfizer-BioNTech BNT162b2 (76.9%) and Moderna Mrna-1273 (23%), followed by CoronaVac and Oxford-AstraZeneca ChAdOx1 nCoV-19. Assessments were conducted predominantly after the second dose and following booster doses (≥3 doses). Additional characteristics including comorbidities, analytical methods used to quantify cytokine concentrations, and further details of the immune response are summarized in Table 1.

### 3.3. Humoral and Cellular Immune Response

The humoral immune response was evaluated by measuring SARS-CoV-2–specific IgG antibodies, primarily those targeting the Spike glycoprotein. Thirteen studies involving older adults were included. In 61.5% of the studies [12,13,14,19,21,22,24,25], IgG titers increased after vaccination, while 23.1% [11,23,27] reported a decrease and 15.4% did not show a significant change [20,26]. Simultaneously, the cellular immune response was evaluated by activating T cells and the secretion of cytokines such as IFN-γ, TNF-α, and various interleukins. Low levels of IFN-γ were reported in 46.2% of the studies [11,20,23,24,26,27], while 38.5% observed an increase after vaccination [13,14,19,21,25].

#### 3.3.1. Humoral Immune Response

Dalla et al. [13] also reported a significant increase in IgG antibodies after the second dose of the Pfizer-BioNTech BNT162b2 vaccine. Additionally, it was observed that 15 days after administration of the booster dose, IgG titers increased up to tenfold. Saiag et al. [14] evaluated the effect of a fourth dose of the Pfizer-BioNTech BNT162b2 vaccine in older adults. The results showed at least a transient increase in anti-Spike antibody levels, challenging the hypothesis that repeated booster doses may induce immune senescence. A significant decrease in anti-S IgG titers over time was also observed. Fukushima et al. [11] administered two doses of mRNA vaccines and found that IgG levels progressively declined with age, and over time since vaccination, in older adults. Dudley et al. [23] observed lower levels of IgG and neutralizing antibodies after the second mRNA vaccine dose, along with a negative correlation between age and anti-spike IgG titers. These findings suggest that lower antibody levels were associated with advanced age, disease status, and medication use. Segato et al. [26] showed reduced IgG levels against JN.1 compared to the ancestral strain following administration of the Pfizer-BioNTech BNT162b2 booster dose. Niyomnaitham et al. [25] reported that with Pfizer-BioNTech BNT162b2 boosters, neutralizing antibody titers against the original Wuhan strain were significantly lower in frail older adults compared to non-frail individuals, while the response to Omicron BA.4/5 was comparable. Rouers et al. [21] demonstrated that, 28 days after a heterologous mRNA vaccine booster dose (BNT162b2 + BNT162b2 + mRNA-1273), higher levels of neutralizing antibodies were observed against the Wuhan and Delta strains. In addition, there was an increase in IgG memory B cells, particularly against the Omicron variant. Kometani et al. [24] found that boosters with Pfizer-BioNTech BNT162b2 vaccines improved humoral immunity by stimulating memory B cells, although cytotoxic activity of CD8+ T cells remained low. Chaiwong et al. [22], evaluating a heterologous CoronaVac/ChAdOx1 regimen in older adults with chronic obstructive pulmonary disease, found neutralizing antibodies against the Wuhan, Alpha, Beta, and Delta variants, but a weak response against Omicron. Costa et al. [20], in their controlled clinical trial with the CoronaVac vaccine, found that IgG levels remained stable against all variants for up to one year. However, no significant increase in IgG response against the SARS-CoV-2 N protein was detected among vaccinated older adults. Bredholt et al. [12] observed an increase in neutralizing antibody titers, mainly after a booster dose with Pfizer-BioNTech, although without improvements in cellular immunity, suggesting a stronger effect on the humoral response. Tut et al. [19] reported that, after the first dose of Pfizer-BioNTech or Oxford-AstraZeneca in older adults without prior SARS-CoV-2 infection, only moderate inhibition was achieved against the viral variants (B.1.1.7, B.1.351, and P.1), reflecting a reduced humoral response in this population. Additionally, they exhibited a delayed antibody response to the first vaccine dose.

#### 3.3.2. Cellular Immune Response

Dalla et al. [13] documented a significant increase in T-cell response after the second Pfizer-BioNTech dose. Additionally, a marked increase in T-cell–specific immune response was observed 15 days after the booster dose. Saiag et al. [14] described an increase in CD4+ T cells producing IFN-γ and TNF-α three weeks after a fourth dose of Pfizer-BioNTech, suggesting a benefit of additional boosters. 

Rouers et al. [21] reported an increase in circulating Th1, Th2, Th17, and follicular helper T-cell responses in older adults who received the heterologous mRNA vaccine booster regimen. 

Costa et al. [20] found that cellular immune response increased during the first 180 days after the first CoronaVac dose, reflected in greater expression of CD40L+ follicular T cells, but this gradually declined thereafter. 

Chaiwong et al. [22] described a growing induction of CD4+ T cells producing TNF-α, IFN-γ, IL-4, IL-17, IL-10, and FasL four weeks after the heterologous CoronaVac/ChAdOx1 regimen, although the response to Omicron was limited. Kometani et al. [24] found that older adults produced fewer circulating follicular helper T cells specific to spike-1. However, these responses improved following the booster dose. Niyomnaitham et al. [25] found that cellular immune response specific to the Spike protein were similar between frail and non-frail individuals, suggesting that frailty may have less impact on cellular immunity. Bredholt et al. [12] observed a lack of strong cytokine-mediated cellular responses after the third Pfizer-BioNTech BNT162b2 dose in older adults, which could explain their vulnerability to severe infections, and may be a consequence of exhausted or senescent cellular immune response. Dudley et al. [23] reported that advanced age was negatively correlated with IL-2 and IFN-γ, along with reduced cellular immune response. Vanda et al. [27] reported reduced IFN-γ levels and a lower proportion of IFN-γ–secreting CD4+ T cells. Fukushima et al. [11] evidenced a progressive decline in IFN-γ with age, particularly in adults over 70 years old, indicating age-related immunosenescence in long-term immunity following vaccination. Tut et al. [19] observed that, despite a limited humoral response, older adults achieved a cellular response comparable to that of younger adults after initial vaccination with Pfizer-BioNTech or Oxford-AstraZeneca.

Collectively, these findings suggest that while humoral immunity tends to increase in older adults after COVID-19 vaccination, cellular immune responses are often diminished, reflecting age-related immunosenescence that may compromise long-term protection in this population. Table 2 summarizes the findings of individual studies.

Appendix A provides a comparative overview of the included studies, categorizing them by vaccine type (mRNA, viral vector, inactivated), population health status (e.g., healthy, frail, or with comorbidities), and primary outcome measures assessed (e.g., cellular response, humoral response). This classification allows for a more nuanced understanding of the heterogeneity among studies, and aids in identifying patterns associated with immunogenicity outcomes across diverse subgroups.

In our review, most studies assessed immune responses between 14 days and 3 months after COVID-19 vaccination. Few studies evaluated these responses beyond 6 months. Only two studies reported that the immune response primarily following booster doses was maintained for more than 6 months in terms of humoral immunity [20,24]. Appendix A provides a detailed overview of the timing of immune response assessment and the corresponding findings.

### 3.4. Sensitivity Analyses

A sensitivity analysis was conducted, considering the exclusion of studies with higher heterogeneity, changes in the statistical model, and the results of the risk-of-bias assessment, for both humoral and cellular immune responses. In the meta-analysis of three studies [14,19,25] that evaluated the humoral immune response using continuous IgG levels (Appendix A), the inverse-variance method with a random-effects model was applied. In addition, the data were presented as mean differences (MDs) with 95% confidence intervals (CIs) and statistical significance of *p* < 0.05. The graphs were designed using Review Manager software version 5.4. The results showed high heterogeneity (MD: 494.42; 95% CI: −383.81 to 1372.64; *p* = 0.27; I^2^ = 98%) (Appendix A). A sensitivity analysis was performed excluding the study contributing the most to heterogeneity [19], and heterogeneity persisted and even increased (I^2^ = 99%; *p* = 0.27), with no significant changes in the overall effect (Appendix A). Subsequently, another sensitivity analysis was performed by changing the statistical model from random-effects to fixed-effects. Although heterogeneity remained high (I^2^ = 98%), a statistically significant difference in the overall effect was observed (*p* < 0.00001) (Appendix A). These findings suggest that the meta-analysis results are sensitive to the statistical model used, which should be considered when interpreting the validity of the estimates. The persistence of heterogeneity suggests the potential influence of other outcomes, such as differences in population characteristics (e.g., age-group stratification), vaccine type used, or the methodology employed in the included studies.

On the other hand, in the meta-analysis of three studies [14,19,25] that evaluated the cellular immune response using continuous IFN-γ levels (Appendix A), the inverse-variance method with a random-effects model was also used. High heterogeneity was found (I^2^ = 98%), and a significant reduction in cellular immune response in older adults was observed (MD: 366.60; 95% CI: 8.59 to 724.62; *p* = 0.04) (Appendix A). A sensitivity analysis was conducted, excluding the study with the greatest heterogeneity [25]. As a result, heterogeneity was moderately reduced (I^2^ = 85%; *p* = 0.0002), with no significant changes in the overall effect (Appendix A).

### 3.5. Risk of Bias and Quality of the Studies

The 13 included studies were assessed using the RoB 2.0 tool and the NOS. Two randomized clinical trials (Costa et al. [20] and Niyomnaitham et al., 2024 [25]) showed a low risk of bias; however, both lacked information regarding allocation concealment (Figure A2). Additionally, 11 observational cohort studies were analyzed. For their evaluation, cutoff points were established: a score ≥ 7 was considered indicative of low risk of bias; a score between 4 and 6 indicated high risk of bias; and a score < 4 indicated very high risk of bias. Overall, all observational studies received scores classifying them as low risk of bias, according to NOS (Table 3).

## 4. Discussion

This systematic review provides a comprehensive synthesis and analysis of humoral and cellular immune responses to COVID-19 vaccines in older adults. Our findings demonstrate increased titers of SARS-CoV-2 Spike-specific IgG antibodies in most of the included studies. This response was particularly evident after the administration of mRNA-based vaccines, notably Pfizer-BioNTech 162b2 and Moderna (mRNA-1273), both after the second dose and booster regimens. On the other hand, cellular immune responses were heterogeneous, and often reduced. Although some studies reported increased production of cytokines such as IFN-γ and TNF-α, particularly in CD4+ T cells, a substantial proportion of studies showed decreased cellular responses. These findings suggest that immunosenescence primarily affects cellular immunity as age progresses. Additionally, factors such as frailty, cognitive impairment, and comorbidities can modulate the magnitude of the immune response.

SARS-CoV-2 infection led to the ongoing pandemic of COVID-19. To control the spread of the disease, various vaccines were developed using different technological platforms [28]. For older adults, the World Health Organization has particularly recommended mRNA-based vaccines, nonreplicating viral vectors, protein subunits, and inactivated virus vaccines [7]. Recent studies have shown that COVID-19 vaccines are effective in containing disease transmission in older adults, with mRNA vaccines demonstrating the highest efficacy [7,29,30]. On the other hand, the effectiveness of the vaccine in reducing hospitalization and admissions to the intensive care unit among older adults has been lower compared to younger age groups [7]. Nonetheless, vaccines have induced significant seroconversion in the older adults, with antibody titers increasing proportionally with the number of doses administered [7,29,30]. However, some studies reported lower antibody titers in older adults compared to younger individuals, likely due to immunosenescence [7,30]. In our review, we observed an increase in the humoral immune response, reflected in higher levels of neutralizing IgG antibodies. In several studies, sustained antibody levels were maintained up to six months after vaccination. Although the cellular immune response was generally reduced, some studies reported increased T-cell activation, highlighting the importance of more precise evaluations of lymphocyte activation in this vulnerable population. These findings emphasize the need to address waning immunity and maintain long-term vaccine efficacy, supporting the recommendation for a booster dose (third dose) to increase antibody levels and provide durable protection in older adults [31].

B cells, CD4+ T cells, and CD8+ T cells play a coordinated role in the immune response against SARS-CoV-2, being essential for both humoral and cellular immunity. Over time, levels of neutralizing antibodies (NAbs) and B cell memory tend to decline, requiring adequate cooperation with T cells to maintain immune protection [32]. Immunoglobulin G, by binding to the receptor-binding domain (RBD) of the viral spike glycoprotein, blocks its interaction with the ACE2 receptor on host cells, thereby neutralizing the virus [32]. IgG also serves as a useful marker for evaluating disease progression and vaccine efficacy. Higher levels of NAbs have been associated with greater immune protection, by reducing viral infectivity. Regarding the cellular response, IFN-γ, secreted by Th1 cells, regulates inflammation, activates macrophages, and promotes the elimination of pathogens. A deficiency in IFN-γ can delay the immune response and facilitate uncontrolled viral replication. Similarly, elevated levels of TNF-α, a key proinflammatory cytokine, are associated with fever, vascular damage, pulmonary complications, and cytokine storm phenomena, which may be linked to a worse prognosis in SARS-CoV-2 infection [32,33,34].

During aging, the immune system undergoes significant changes, known as immunosenescence, which affect both innate and adaptive immunity. This immunological remodeling plays a major role in the pathogenesis of various chronic diseases in older adults, particularly in conditions such as neurodegenerative disorders, cancer, cardiovascular diseases, autoimmune disorders, and COVID-19 [9,35]. Recent studies have shown a decline in the production of precursor B cells in the bone marrow of both aged mice and humans [36]. These changes are partly attributable to advanced age, and are associated with reduced levels of cytokines such as IL-17, critical for B-cell survival, as well as a greater bias of hematopoietic stem cells toward myeloid differentiation at the expense of lymphopoiesis [36]. Germinal centers, essential for antibody-affinity maturation, require functional interactions with follicular helper T cells and follicular regulatory T cells. In animal models, age-related changes in CD4+ T cells have been shown to contribute to defective germinal center responses and reduced antibody production following immunization [37]. Moreover, naïve T cells, key to generating memory T cells during primary immune responses, decline in absolute number with age [38]. While the number of CD4+ naïve T cells tends to remain relatively sufficient in healthy older adults, CD8+ naïve T cells are more severely affected, to the extent that the size of the CD8+ naïve T-cell repertoire may become limiting for generating effective memory responses [39]. Four major subsets of memory T cells have been described in the literature: naïve T cells, central memory T cells, effector memory T cells, and terminally differentiated effector T cells. These subsets can be characterized by the expression of specific surface markers, including CCR7, CD45RA/RO, CD27, CD28, CD62L, and CD95. Although the total number of T cells remains relatively stable with aging, significant changes have been observed in the composition of memory subpopulations, particularly in CD4+ and CD8+ naïve T cells [40]. Finally, pro-inflammatory cytokines play a central role in age-related immune remodeling. The sustained increase in their production during aging has been linked to multiple mechanisms, including cellular senescence, mitochondrial dysfunction, DNA damage, and alterations in gut microbiota composition. These processes contribute to a chronic low-grade inflammatory state, known as inflammaging, which negatively affects the regulation and effectiveness of the adaptive immune response [41].

Immunosenescence, defined as the gradual deterioration of the immune system with aging, affects both humoral and cellular responses. It is associated with reduced production of B and T lymphocytes, due to structural and functional changes in the bone marrow and thymus, respectively [36]. Thymic involution, marked by the progressive loss of cortical and medullary epithelial tissue, leads to a dramatic decline in the output of naive T cells [42]. This limits the generation of immune responses to novel antigens such as SARS-CoV-2 [43]. Concurrently, dysfunction in germinal centers within lymph nodes and the expansion of senescent effector T-cell populations diminish the quality of the adaptive immune response [44]. These alterations contribute to reduced vaccine efficacy, even when antibody titers are present [45]. However, booster doses have been shown to significantly enhance immunogenicity in older adults, increasing not only the quantity, but also the functional quality of antibodies, and enhancing T-cell activation [7,13]. However, this response remains heterogeneous, and may depend on baseline immune status, the presence of chronic comorbidities, and the degree of individual immunosenescence [46]. These findings underscore the critical role of booster doses in the older adults, as both humoral and cellular responses have been shown to improve after their administration.

The limited efficacy of the vaccine in this age group is influenced not only by biological aging factors, but also by social and behavioral elements [7]. Distrust towards new vaccine technologies, skepticism about their efficacy, and concerns about side effects have contributed to lower acceptance and coverage among older adults [7]. This poses an additional challenge to achieving population-level immunity in this vulnerable group. Therefore, strategies should be implemented to enhance vaccination coverage, such as leveraging information and communication technologies, which may be effective tools in reaching older adults [47].

The most relevant promotional strategies include free vaccination, educational materials, posters and reminder messages, as well as communication through emails, audiovisual resources community counseling, and physician recommendations, using informative brochures [48]. To optimize the effectiveness of COVID-19 immunization in older adults, heterologous vaccination has been employed, as it provides broader protection, higher antibody levels, and a more comprehensive immune response [49]. Early administration of booster doses is also recommended, to sustain the immunity acquired from primary vaccination. In addition, continuous monitoring of the immune response elicited by vaccination and booster doses is essential [29,49].

Comparisons with other age groups reinforce these observations. In the pediatric population, for example, SARS-CoV-2 infection has presented mostly with mild symptoms, yet the vaccine-induced immune response has been robust [50,51,52]. A cohort study in China showed that after two doses of inactivated vaccines, seroconversion was nearly universal in children, and significant activation of CD4+ and CD8+ T cells was observed [52]. Most adverse events in this population were mild, although isolated cases of myocarditis and pericarditis were reported [50,51]. In young adults, the immune response after vaccination has been superior in both magnitude and duration, for humoral and cellular components alike [53]. This difference may be due not only to the absence of immunosenescence, but also to higher baseline immunocompetence [54]. A comparative study that evaluated immune responses after two CoronaVac boosters found that young adults exhibited significantly higher levels of pro-inflammatory cytokines such as TNF-α, whereas older adults showed a predominance of regulatory cytokines such as IL-10 and IL-2, suggesting a more modulated immune response potentially shaped by immunological aging [55]. This pattern likely results from both the aging of T cells and an increased proportion of regulatory T-cell subsets in older individuals [55]. Regarding safety, vaccine-related adverse events were reported more frequently in young adults than in older adults [30]. In a cohort study evaluating immune responses in young adults following administration of a third (mRNA-based) vaccine dose, higher levels of humoral immunity measured by anti-Spike IgG and NAbs were observed in vaccinated individuals with prior SARS-CoV-2 infection (hybrid immunity), compared to vaccinated individuals without prior infection [56]. However, no significant difference was found in cellular immune responses, as measured by IFN-γ, before and after vaccination. Nevertheless, the third dose significantly enhanced both humoral and cellular immune responses [56]. In contrast, a comparative study between young and older adults demonstrated that, following vaccination, serum-neutralizing antibody levels, IgG titers, IFN-γ, and interleukin-2 were all higher in younger individuals, indicating an age-associated decline in acquired immune responses beyond 80 years of age [57]. Similarly, another study showed that IgG titers and post-vaccination cellular immune responses after initial mRNA vaccination were significantly higher in younger adults compared to older adults [58].

Our findings support the notion that aging may influence the immune response to COVID-19 vaccines. However, as Fulop et al. [59] highlight, the association between immunosenescence and reduced vaccine efficacy is not absolute. Their review underscores the fact that recent advances in vaccine design, such as the development of effective formulations for herpes zoster and SARS-CoV-2, have enabled comparable protection in older and younger individuals. Thus, while immunosenescence remains a relevant factor, it is essential to consider other variables such as chronic conditions, prior immunity, and individual variability, which may modulate vaccine responses in older adults.

The studies included in this review exhibited considerable methodological variability in terms of vaccine platform, timing of immune response assessment, and assay methods used. Such heterogeneity may significantly influence the magnitude and duration of immune responses reported. For example, studies assessing immune parameters within 1 month post-vaccination often reported higher responses compared to those with longer follow-up periods. Similarly, differences in assay sensitivity and specificity (e.g., ELISpot vs. flow cytometry for cellular responses) can affect the comparability of findings.

The decline in cellular immune response among older adults may be influenced by multiple factors not addressed in the present review, such as a history of prior infections, epidemic or pandemic outbreaks, the emergence of new viral variants, predisposing genetic mutations, and the presence of multiple comorbidities or conditions such as frailty, which can contribute to an altered cellular immune environment. This reduction in cellular immunity may have important clinical implications, including increased susceptibility to breakthrough infections following vaccination, a higher risk of developing severe COVID-19, and a shorter duration of long-term immune protection.

Although the inclusion criterion was ≥60 years, it is important to recognize that cellular immune responses may vary significantly across age subgroups, such as adults aged 60 to 70 and those over 80. This distinction is relevant, as the progressive aging of the immune system can differentially affect both the magnitude and duration of cellular and humoral immune responses. However, the studies included in this review did not perform stratified analyses by advanced age groups, limiting the ability to provide more specific clinical recommendations for the most vulnerable populations. Moreover, although this review focused on cellular immune parameters, the direct relationship between these indicators and clinical outcomes such as post-vaccination infections, severe disease, or hospitalization was not addressed in the primary studies. This limitation reduces the immediate clinical applicability of the findings and highlights the need for future research that integrates immunological data with concrete clinical outcomes, to strengthen their relevance in medical practice.

It is recommended that future studies employ standardized and comparable designs, incorporating longitudinal measurements to assess the persistence and quality of the immune response over time in older adults. Greater emphasis should be placed on using narrower age stratifications (e.g., 60–69, 70–79, 80–89, and ≥90 years) and accounting for clinically relevant variables such as comorbidities, frailty, and indicators of muscle health (e.g., dynapenia or sarcopenia). Additionally, the development of personalized vaccination strategies aimed at optimizing both humoral and cellular immune responses in the elderly should be prioritized.

Our review has several limitations. First, most of the included studies were observational, due to the limited availability of clinical trials specifically evaluating humoral and cellular immune responses in older adults. Furthermore, only studies that evaluated both responses simultaneously were considered, while those that evaluated them separately were excluded. Second, the sample sizes in the included studies were relatively small, which may limit the generalizability of the findings. Third, methodological heterogeneity across studies, in terms of the techniques used to measure immune responses and the baseline clinical characteristics of participants, made direct comparisons and generalizations challenging. Fourth, an important limitation was the broad age grouping used in the included studies. Although the aging process is associated with progressive immunological changes, many of the primary studies did not stratify participants into specific age subgroups. This lack of categorization prevented a more detailed sub analysis of the effect of immunosenescence, according to increasing age. As a result, the findings may not accurately reflect the subtle immunological differences that occur within the older adult population, limiting the ability to identify specific patterns of vaccine response or immunological vulnerability associated with more advanced age groups. Fifth, it was not possible to perform a subgroup analysis by vaccine platform for continuous outcomes, as all eligible studies reporting adequate statistical measures focused exclusively on mRNA-based vaccines. This limitation restricts the ability to compare the immune response between different vaccine technologies. Finally, some studies did not provide complete information on comorbidities, functional status, or frailty, all of which may significantly influence the magnitude and quality of the immune response.

## 5. Conclusions

Older adults who receive COVID-19 vaccination, particularly booster regimens and mRNA-based vaccines, exhibit enhanced humoral immunity but reduced cellular responses. These findings reflect age-related immunosenescence, which may compromise long-term immunity in this population. Furthermore, they highlight the importance of developing vaccination strategies that both effectively stimulate antibody production and also optimize T-cell activation in this vulnerable group.

## Figures and Tables

**Table 1 vaccines-13-00852-t001:** Characteristics of studies include humoral and cellular immune response.

First Author Name/Year of Publication	Country	Design	Period	Total, Sample n	Age—Older Adults	Antecedents—Comorbidities	Sample Immune Response	Masculine n (%)	Vaccine Type (Technology)	Dose	Test or Assay to Evaluate CIR	Test or Assay to Evaluate HIR
Tut et al., 2021 [19]	UK	Cohort	11 December 2020–16 February 2021	35	≥65	None	35	NR	Pfizer-BioNTech (mRNA) Oxford-AstraZeneca (viral vector)	1st	Human IFN-γ ELISpotPRO	V-PLEX SARS-CoV-2 Panel 2
Costa et al., 2022 [20]	Brazil	Controlled clinical trial	July 2020—NR	24	≥60	None	24	16 (66.7)	CoronaVac (Inactivated)	2nd	AIM/flow cytometry	Electrochemiluminescence multiplex serology assay
Rouers et al., 2022 [21]	Singapore	Cohort	NR	41	≥60	None	41	20 (48.7)	Pfizer-BioNTech (mRNA) Moderna (mRNA)	2nd	ICS Multicolored flow cytometry	ELISpot Flow Cytometry Pseudovirus neutralization
Chaiwong et al., 2023 [22]	Thailand	Cohort	July 2021 and January 2022	30	≥65	COPD	30	25 (83.3)	CoronaVac (Inactivated) Oxford—AstraZeneca (viral vector)	2nd	BD FACSCelestaTM flow cytometer	cPass SARS-CoV-2
Dalla et al., 2023 [13]	Italy	Cohort	October 2021–January 2022	49	≥70	Dementia CVD Diabetes COPD Autoimmune disease	49	12 (24.4)	Pfizer-BioNTech (mRNA)	3rd	ELISpot	ELISA Competitive ELISA for Nab
Dudley et al., 2023 [23]	USA	Cohort	February 2021–January 2022	46	≥60	RA HBP Diabetes CAD	46	37 (80)	Pfizer-BioNTech (mRNA) Moderna (mRNA)	3rd	ELISPOT	Bead multiplex immunoassay Pseudovirus neutralization assay
Saiag et al., 2023 [14]	Israel	Cohort	August 2021–January 2022	133	≥60	None	133	50 (37.5)	Pfizer-BioNTech (mRNA)	4th	SARS-CoV-2 T-Cell Analysis Kits for human PBMCs	ADVIA Centaur SARS-CoV-2 IgG CMIA
Bredholt et al., 2024 [12]	Norway	Cohort	NR	68	≥70	CHD CLD RD Diabetes Cancer Immunosuppression CKD Neurological disease	68	29 (43)	Pfizer–BioNTech (mRNA) Moderna (mRNA)	3rd	FluoroSpot ELISpot	ELISA Microneutralization
Kometani et al., 2024 [24]	Japan	Cohort	NR	105	≥65	None	105	55 (50.4)	Pfizer-BioNTech (mRNA)	3rd	AIM	Surrogate Virus Neutralization Assay Flow cytometry
Niyomnaitham et al., 2024 [25]	Thailand	Randomized clinical trial	9 January–8 August 2022	139	≥65	HBP Dyslipidemia DM	139	45 (32.37)	Pfizer-BioNTech (mRNA) Oxford-AstraZeneca (viral vector)	2nd	ELISpot	CMIA Neutralization assay
Segato et al., 2024 [26]	Italy	Cohort	January 2024–15 March 2024	18	≥65	Diabetes COPD CVD CRF	18	12 (66.6)	Pfizer-BioNTech (mRNA)	≥3rd	ELISpot AIM	ELISA
Vanda et al., 2024 [27]	Singapore	Cohort	NR	14	≥70	None	14	NR	Pfizer-BioNTech (mRNA)	2nd	ICS Flow cytometry	High-dimensional flow cytometry
Fukushima et al., 2025 [11]	Japan	Cohort	September 2021–March 2022	80	≥70	None	80	9 (11.2)	mRNA	2nd	QuantiFERON	SARS-CoV-2 IgG II Quant

AIM: Activation-induced marker assay; CHD: chronic heart disease; CIR: cellular immune response; CLD: chronic lung disease; CMIA: chemiluminescent microparticle immunoassay; COPD: chronic obstructive pulmonary disease; CRF: chronic renal failure; CVD: cardiovascular disease; DM: diabetes mellitus; ECLIA: electrochemiluminescence sandwich immunoassay; ELISA: enzyme-linked immunosorbent assay; ELISpot: enzyme-linked ImmunoSpot; GID: gastrointestinal disease; HIR: humoral immune response; ICS: intracellular cytokine staining; ID: immunodeficiency or immunosuppression; IFN-γ: interferon gamma; IgG: Immunoglobulin G; mRNA: messenger ribonucleic acid; Nab: neutralizing antibodies; NR: not reported; RA: rheumatoid arthritis; PBMCs: peripheral blood mononuclear cells; RD: rheumatic disease; RLC: severe renal, lung or cardiac disease; SARS-CoV-2: severe acute respiratory syndrome coronavirus 2; UK: United Kingdom.

**Table 2 vaccines-13-00852-t002:** Humoral and cell response of COVID-19 Vaccine among older adults.

First Author Name/Year of Publication	Humoral Immune Response	Findings HIR	Cellular Immune Response	Findings CIR	Results	Overall Conclusions
Tut et al., 2021 [19]	↑ IgG levels	Moderate inhibition against B.1.1.7, B.1.351, and P.1 variants after the first dose of Pfizer-BioNTech or AstraZeneca; reduced humoral response.	↑ IFN-γ	Cellular response comparable to young adults after first dose of Pfizer or AstraZeneca.	No correlation was observed between IgG and IFN-γ values.	HIR decreases in older adults without prior infection. CIR is maintained under pre-infection conditions.
Costa et al., 2022 [20]	IgG levels are maintained	Stable IgG levels over one year with CoronaVac; no significant increase against N protein.	↓ IFN-γ levels	Increase in CD40L+ T cells during first 180 days with CoronaVac; decline thereafter.	HIR is maintained and the CIR decreases; no correlation between IgG and IFN-γ.	The HIR is maintained and the CIR decreases in older adults.
Rouers et al., 2022 [21]	↑ IgG levels	Higher levels of neutralizing antibodies and IgG memory B cells after heterologous mRNA booster.	↑ IFN-γ levels	Increase in Th1, Th2, Th17, and follicular helper T cells after heterologous mRNA booster.	HIR and CIR are enhanced by heterologous boosters in the elderly.	CIR and HIR increase in older adults after the heterologous mRNA booster vaccine.
Chaiwong et al., 2023 [22]	↑ IgG anti-S1 levels	Neutralizing antibodies detected against Wuhan, Alpha, Beta, and Delta variants; weak response to Omicron.	Limited IFN-γ and TNF-α response	Induction of CD4+ TNF-α, IFN-γ, IL-4, IL-17, IL-10, and FasL at 4 weeks after CoronaVac/ChAdOx1 schedule.	Humoral response against the Alpha, Beta, and Delta variants, and a limited cellular response.	CIR limited and HIR increases in older adults.
Dalla et al., 2023 [13]	↑ IgG levels	Significant IgG increase after the second dose and Pfizer-BioNTech booster.	↑ IFN-γ levels	Significant increase in T-cell response after second dose and Pfizer booster.	HIR and CIR increase after booster vaccine; no correlation between IgG and IFN-γ values.	CIR and HIR increase in older adult patients after BNT162b2 booster.
Dudley et al., 2023 [23]	↓ IgG levels	Low IgG and neutralizing antibody levels after second dose; negative correlation with age.	↓ IFN-γ levels	Reduced IL-2, IFN-γ, and T-cell response in older adults.	A correlation was observed between IgG and IFN-γ levels.	CIR and HIR decreases in older adult patients.
Saiag et al., 2023 [14]	↑ IgG levels	Transient increase in anti-Spike antibodies after the fourth dose; progressive IgG decline.	↑ IFN-γ and TNF-α	Increase in CD4+ IFN-γ and TNF-α after the fourth Pfizer dose.	The fourth dose of vaccine significantly enhanced both humoral and cell reactivity.	CIR and HIR increase in older adult patients.
Bredholt et al., 2024 [12]	↑ IgG levels	Increased neutralizing antibodies after Pfizer-BioNTech booster; no improvement in cellular immunity.	There is no significant improvement in T-cell immunity.	No robust cellular response after third dose; possible T-cell exhaustion.	Lower humoral and cell responses in older adults.	Booster dose improves HIR and CIR decreases in older adult patients.
Kometani et al., 2024 [24]	↑ IgG levels	Pfizer booster stimulated memory B cells; low CD8+ T-cell response.	↓ Cells T CD8+ and cTfh1	Reduced production of follicular helper T cells; improved after booster.	HIR increase after mRNA booster; no correlation between IgG levels and T cells.	HIR increase in older adults, while the CIR decreases with age.
Niyomnaitham et al., 2024 [25]	↑ IgG levels	Lower antibody titers against Wuhan strain in frail older adults; comparable Omicron response.	↑ IFN-γ	Spike-specific T-cell response similar between frail and non-frail participants.	Reduced IgG antibody levels in the frail IM group; similar IFN-γ secretion in both groups.	CIR and HIR increase in older adult patients.
Segato et al., 2024 [26]	IgG levels are maintained	Reduced IgG against JN.1 after Pfizer booster.	↓ IFN-γ levels	Lower CD4+ responses against JN.1 and BA.2.86 after Pfizer booster.	IgG antibody levels were uniform and levels of IFN-γ were reduced.	CIR decreases in older adult patients.
Vanda et al., 2024 [27]	↓ IgG levels	It focused mainly on T cells.	↓ IFN-γ and TNFα levels	Reduced IFN-γ levels and lower proportion of IFN-γ–secreting CD4+ T cells.	HIR and CIR are reduced in older adults; correlation between IgG and IFN-γ.	HIR and CIR are reduced in older adults.
Fukushima et al., 2025 [11]	↓ IgG levels	Progressive decline in IgG levels with age and time after mRNA vaccination.	↓ IFN-γ levels	Progressive IFN-γ decline with age in individuals over 70.	No correlation was observed between IgG antibody levels and secreted IFN-γ values.	CIR and HIR decreases in older adult patients.

CIR: cellular immune response; HIR: humoral immune response; IFN-γ: interferon-gamma; IgG: immunoglobulin G; TNF-α: tumor necrosis factor-alpha. ↑: increase levels; ↓: decrease levels.

**Table 3 vaccines-13-00852-t003:** Risk-of-bias and quality assessment using the NOS tool.

First Author/Year of Publication	Selection	Comparability	Exposure/Outcome	Score
Tut et al., 2021 [19]	****	*	***	8
Rouers et al., 2022 [21]	***	*	***	7
Chaiwong et al., 2023 [22]	***	*	***	7
Dalla et al., 2023 [13]	****	*	**	7
Dudley et al., 2023 [23]	****	*	**	7
Bredholt et al., 2024 [12]	***	*	***	7
Saiag et al., 2023 [14]	***	*	***	7
Kometani et al., 2024 [24]	****	*	**	7
Segato et al., 2024 [26]	****	*	***	8
Vanda et al., 2024 [27]	****	*	**	7
Fukushima et al., 2025 [11]	***	*	***	7

* NOS scale score value.

## Data Availability

Additional data related to this paper may be requested from the authors.

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
