# Peer review of "Humoral and Cellular Immune Responses Against SARS-CoV-2 Following COVID-19 Vaccination in Older Adults: A Systematic Review"

_vaccines, 2025, doi:10.3390/vaccines13080852_

Round 1
Reviewer 1 Report
Comments and Suggestions for Authors
In this Systematic Review, Rojas-De la cruz et al. search for studies that reported antibody and T cell responses to SARS-CoV2 vaccination in older adults. While the number of studies included is small, this is probably due to SARS-CoV2 being a relatively new virus (5ish years) and getting a population of older adults for the original research can be difficult. The highlight of this review is the compilation of data from the original studies in 2 tables. While systematic reviews in MDPI journals require specifics in “methods” (vs. a regular review) are required, the bulk of the manuscript should still be the discussion of the findings from original research. Additionally, this manuscript is hard to review and provide specific comments since no line numbers were provided on this reviewer’s copy of the manuscript, which are provided in the mpdi template.
General comments:
-When resubmitting, please be sure to add line numbers in the manuscript, as well a reference them (including track changes so reviewers can see/follow) with applicable changes.
-Section 3 (which the summary of the findings) is the heart of this review, particularly Section 3.3 (the review of the published original research findings/data). While this section summarizes findings in a concise matter, this section should be at least 50% of the review manuscript, when the text 1 page of 10 (the key tables mentioned in the summary are key, and do take another 2 pages). A few specific article findings are mentioned, but overall the section is too summarized (levels go up, then down, etc.) and should be expanded with more of the findings from the different studies (instead of solely relying on the table(s). This could even be separated into 2 subsections (antibodies and t cells). To not make the manuscript longer, the discussion should be shortened and possibly the methods could be shortened and/or more moved to supplemental.
Specific Comments:
-Article type must be changed, as “Articles” are for original research (which this is not) as outlined on mdpi’s website on types of manuscripts. This is a review (if the editor wants Systematic Review as the type, that is ok-as the mpdi website does list this as a separate type).
-Last paragraph of introduction: Considering this is a newer disease (barely 5 years), it make sense that there are few long-term studies. This sentence should be rewritten for relevance of the current status/only have 5 years of data.
-3.3 First paragraph: Citations are needed (or figure/table reference to see this combination/number and how this was calculated).
-3.3 Second paragraph/Kometanti et al[24]: Which mRNA vaccines were in this study? Earlier brands were mentioned, and different brands were known to have different amounts of mRNA.
Author Response
Dear Editor and Reviewers,
Thank you for your valuable suggestions and observations. Below, we provide a detailed response to each
of the comments.
Reviewer 1
General comments:
Line numbers have been added, and all changes made in response to the reviewers’ suggestions have been
highlighted.
Section 3 was restructured into Cellular Immune Response and Humoral Immune Response. Additionally,
Table 2 was expanded with further information. The depth of analysis has been increased without
substantially extending the length of the manuscript (Lines 215–286).
Specific comments:
The term "Article" was changed to "Systematic Review" (Line 1).
The final paragraph of the introduction was rewritten (Lines 85–92).
3.3 First paragraph – Citations are needed: References were added to support each analysis (Lines 208–
214).
3.3 Second paragraph / Kometani et al. [24]: The specific vaccine used ("Pfizer BioNTech 162b2") was
added to the description of the Kometani et al. study (Line 238).
Reviewer 2
The references were reviewed, and the corresponding section of the discussion was rewritten (Lines 70–
73). All references in the manuscript were carefully verified.
We agree that incorporating contrasting perspectives is essential for a balanced scientific discussion. A
citation from the suggested article was included. A brief mention was added in the introduction (Lines 81–
84), and the topic was further developed in the discussion section (Lines 422–430).
Reviewer 3
Major comments:
1. We appreciate this valuable comment. A new section was added to the discussion addressing the
suggestion (Lines 439–446).
2, 5, and 6. A complete paragraph was written in the discussion section covering all the
mechanistic aspects requested (Lines 348–378).
2. A detailed explanation was added clarifying why a meta-analysis could not be performed (Lines
159–170).
4 and 10. A paragraph was added in the Results and Discussion sections addressing these points
(Lines 431–438), along with a new summary table (Table S2, Lines 296–301).
8 and 9. A paragraph was added to the discussion addressing these two points (Lines 447–459).
3. A brief methodological appendix was added on heterogeneity (Methodological Appendix, Table
S1) (Line 170).
Minor comments:
1. Terminology was standardized throughout the manuscript using "Older adults".
2. All abbreviations were carefully reviewed and standardized.
3. Numerical discrepancies were verified and corrected for the studies by Kometani et al. [24] and
Segato et al. [26]. These changes do not affect the analyses or results.
4. The suggested correction was implemented (Line 66).
5. The suggested change was made (Line 26).
6 and 8. The reviewer’s suggestion was incorporated (Lines 47–48).
6. Terminology was standardized across the manuscript using "Cellular Immune Response".
7. Complex sentences were simplified, and the suggested revision was incorporated (Lines 418–
420)

Reviewer 2 Report
Comments and Suggestions for Authors
Dear Authors,
The manuscript entitled “Humoral and Cellular Immune Responses Against SARS-CoV-2 Following COVID-19 Vaccination in Older Adults. A Systematic Review” addresses a relevant topic and presents important findings.
The manuscript provides an objective overview of the published studies on the subject and will be a useful source of information for researchers interested in the topic.
I have two comments I would like the authors to consider. First, in every scientific manuscript, the statements made should be supported by references to scientific articles. When analyzing the Introduction, I noticed that in the excerpt “In older adults…levels after vaccination \[3,10],” neither of the cited references are scientific articles containing specific results that support the statement. Therefore, I suggest that the authors carefully review all the references used in the manuscript.
Second, I believe the manuscript would be enriched if the authors included in the Introduction and Discussion studies suggesting that there is not necessarily a direct relationship between reduced vaccine response and senescence (see: [https://doi.org/10.3390/vaccines10040607](https://doi.org/10.3390/vaccines10040607)). Including opposing viewpoints is always important in a scientific article.
I hope these comments are helpful.
Best regards,
Author Response

(The authors gave the same response as above.)

Reviewer 3 Report
Comments and Suggestions for Authors
The manuscript reviews and synthesizes current evidence on humoral and cellular immune responses to COVID-19 vaccination in adults aged 60 years and older. The authors analyzed 13 studies (782 participants) and found that while humoral immunity (IgG antibodies) tends to increase post-vaccination (61.5% of studies), cellular immunity often shows reduced responses (46.2% reported low IFN-γ levels). The central hypothesis addresses age-related immunosenescence compromising vaccine efficacy in older adults. The work contributes meaningfully to understanding vaccine immunogenicity in vulnerable populations and supports tailored vaccination strategies.
Major Comments:
1 The paper concludes that reduced cellular immunity may compromise long-term protection but does not sufficiently discuss the real-world consequences (e.g., breakthrough infection, severe disease risk, or duration of immunity) in the context of waning cellular responses.
2 Although immunosenescence is identified as a key driver, the mechanistic underpinnings (B vs. T cell aging, memory cell function, cytokine milieu changes) are not explored in detail. This could help contextualize findings for a molecular biology audience.
3 The decision to perform only a narrative synthesis due to heterogeneity is appropriate; however, more detail on attempted data harmonization and reasons why meta-analysis was not feasible would strengthen the justification.
4 Consider including a summary table that categorizes studies by vaccine type, population health status, and outcome measures. Explicitly discuss how these variables impact the generalizability of the results.
5 Expand on what reduced cellular responses mean for protection in older adults. Reference studies linking immune parameters to infection or hospitalization risk.
6 Incorporate recent mechanistic studies on immunosenescence, particularly those detailing changes in T cell subpopulations, exhaustion markers, or age-associated B cell dysfunction.
7 Provide a brief methodological appendix or supplementary figure illustrating the heterogeneity (e.g., a forest plot of key outcomes, even if not pooled) to help readers understand the limitations.
8 Insufficient Age Stratification Analysis:While the inclusion criterion is ≥60 years, there's minimal discussion of immune response differences between "young-old" (60-70) and "old-old" (>80) subgroups, which could significantly impact clinical recommendations.
9 Limited Clinical Correlation. The relationship between measured immune parameters and clinical outcomes (breakthrough infections, hospitalization rates) is not adequately addressed, limiting the clinical relevance of findings.
10 Add a dedicated subsection in the Discussion analyzing how methodological differences across studies may influence results. Consider creating a supplementary table categorizing studies by key variables (vaccine type, measurement timing, assay methods).
Minor Comments:
1 "Older adults," "elderly," and "aged individuals" are used interchangeably. Standardize terminology throughout.
2 Some abbreviations are introduced multiple times (e.g., IFN-γ).
3 Some numerical discrepancies need verification (e.g., participant numbers across studies)
4 Page 2, Line 2: Change "progressively immune decline" to "progressive immune decline"
5 Page 8, Abstract section: Consider replacing "Evidence on the humoral and cellular immune responses...remains limited" with "Evidence on the humoral and cellular immune responses...requires systematic synthesis"
6 Page 10: The phrase "may compromise long-term immunity" should be strengthened to "likely compromises" given the evidence presented.
7 Terminology for cellular immunity varies ("CIR," "cellular immune response," "T cell response"); standardize for clarity.
8 Replace: “...which may compromise long-term immunity in this population”
With: “...which may limit the durability and breadth of protection following vaccination in older adults.”
9 Simplify complex sentences, e.g., Original: “This immunological pattern may reflect both T-cell senescence and the expansion of regulatory T cell subsets with age [44].”Suggested: “This pattern likely results from both the aging of T cells and an increased proportion of regulatory T cell subsets in older individuals [44].”
Author Response

(The authors gave the same response as above.)

Round 2
Reviewer 1 Report
Comments and Suggestions for Authors
The authors have addressed all reviewer comments to the best of their ability. This reviewer agrees the manuscript can be published in its current form.
Author Response
Dear Editor and Reviewers,
Thank you for your valuable suggestions and observations. Below, we provide a detailed response to each
of the comments.
Reviewer 1
General comments:
Line numbers have been added, and all changes made in response to the reviewers’ suggestions have been
highlighted.
Section 3 was restructured into Cellular Immune Response and Humoral Immune Response. Additionally,
Table 2 was expanded with further information. The depth of analysis has been increased without
substantially extending the length of the manuscript (Lines 215–286).
Specific comments:
The term "Article" was changed to "Systematic Review" (Line 1).
The final paragraph of the introduction was rewritten (Lines 85–92).
3.3 First paragraph – Citations are needed: References were added to support each analysis (Lines 208–
214).
3.3 Second paragraph / Kometani et al. [24]: The specific vaccine used ("Pfizer BioNTech 162b2") was
added to the description of the Kometani et al. study (Line 238).
Reviewer 2
The references were reviewed, and the corresponding section of the discussion was rewritten (Lines 70–
73). All references in the manuscript were carefully verified.
We agree that incorporating contrasting perspectives is essential for a balanced scientific discussion. A
citation from the suggested article was included. A brief mention was added in the introduction (Lines 81–
84), and the topic was further developed in the discussion section (Lines 422–430).
Reviewer 3
Major comments:
1. We appreciate this valuable comment. A new section was added to the discussion addressing the
suggestion (Lines 439–446).
2, 5, and 6. A complete paragraph was written in the discussion section covering all the
mechanistic aspects requested (Lines 348–378).
2. A detailed explanation was added clarifying why a meta-analysis could not be performed (Lines
159–170).
4 and 10. A paragraph was added in the Results and Discussion sections addressing these points
(Lines 431–438), along with a new summary table (Table S2, Lines 296–301).
8 and 9. A paragraph was added to the discussion addressing these two points (Lines 447–459).
3. A brief methodological appendix was added on heterogeneity (Methodological Appendix, Table
S1) (Line 170).
Minor comments:
1. Terminology was standardized throughout the manuscript using "Older adults".
2. All abbreviations were carefully reviewed and standardized.
3. Numerical discrepancies were verified and corrected for the studies by Kometani et al. [24] and
Segato et al. [26]. These changes do not affect the analyses or results.
4. The suggested correction was implemented (Line 66).
5. The suggested change was made (Line 26).
6 and 8. The reviewer’s suggestion was incorporated (Lines 47–48).
6. Terminology was standardized across the manuscript using "Cellular Immune Response".
7. Complex sentences were simplified, and the suggested revision was incorporated (Lines 418–
420).

Reviewer 3 Report
Comments and Suggestions for Authors
The manuscript systematically reviews humoral and cellular immune responses in individuals aged 60 years and older following COVID-19 vaccination. Thirteen studies (eleven observational, two RCTs) were included, focusing mainly on mRNA vaccines. The review finds that while humoral immunity (e.g., anti-Spike IgG titers) is generally enhanced after vaccination and booster doses, cellular immune responses (such as IFN-γ production) are frequently reduced, reflecting age-related immunosenescence. The study concludes that tailored vaccination strategies are needed to optimize protection in older adults.
Comments
1 Pooling adults aged 60-90+ as a single group oversimplifies the immunosenescence spectrum
2 Critical gap in addressing differential responses across age decades within the "older adult" category
3 Conduct subgroup analyses where possible by vaccine platform (mRNA vs viral vector vs inactivated)
4 Stratify findings by timing of assessment (1-3 months vs >6 months post-vaccination)
5 Consider sensitivity analyses excluding studies with highest heterogeneity
6 Add a dedicated section discussing the limitation of broad age grouping
7 Include recommendations for future studies to employ narrower age stratification
8 Discuss implications for vaccination strategies across different elderly subgroups
9 Expand discussion of correlates of protection and clinical significance of observed immune parameters
10 Include more robust comparison with younger adult responses from the literature
11 Line 22: "requires systematic synthesis" - consider "warrants" for stronger language
12 Line 275: "Taken together, the studies indicate..." - awkward transition; suggest "Collectively, these findings suggest..."
13 Table 1: Several abbreviations lack definition in the legend
14 Line 39: Change "appear reduced" to "are frequently diminished"
15 Line 304: "enhanced humoral response after vaccination, characterized by increased titers" - redundant phrasing
16 Line 468: "not only effectively stimulate" should be "both effectively stimulate"
Author Response

(The authors gave the same response as above.)
